# QUANTIFYING AND ENHANCING MULTI-MODAL RO-BUSTNESS WITH MODALITY PREFERENCE

**Zequn Yang[1], Yake Wei[1], Ce Liang[1], Di Hu[1]***
Gaoling School of Artificial Intelligence, Renmin University of China[1]
{zqyang,yakewei,liangce158,dihu}@ruc.edu.cn

## ABSTRACT

Multi-modal models have shown a promising capability to effectively integrate information from various sources, yet meanwhile, they are found vulnerable to pervasive perturbations, such as uni-modal attacks and missing conditions. To counter these perturbations, robust multi-modal representations are highly expected, which are positioned well away from the discriminative multi-modal decision boundary. In this paper, different from conventional empirical studies, we focus on a commonly used joint multi-modal framework and theoretically discover that larger uni-modal representation margins and more reliable integration for modalities are essential components for achieving higher robustness. This discovery can further explain the limitation of multi-modal robustness and the phenomenon that multi-modal models are often vulnerable to attacks on the specific modality. Moreover, our analysis reveals how the widespread issue, that the model has different preferences for modalities, limits the multi-modal robustness by influencing the essential components and could lead to attacks on the specific modality highly effective. Inspired by our theoretical finding, we introduce a training procedure called *Certifiable Robust Multi-modal Training* (CRMT), which can alleviate this influence from modality preference and explicitly regulate essential components to significantly improve robustness in a certifiable manner. Our method demonstrates substantial improvements in performance and robustness compared with existing methods. Furthermore, our training procedure can be easily extended to enhance other robust training strategies, highlighting its credibility and flexibility.

## 1 INTRODUCTION

As data are often presented from different perspectives, like text, images, and audio, how to effectively exploit and integrate information from multiple sources becomes important. This has given rise to the concept of multi-modal learning, which serves as a potent approach that enables a more comprehensive understanding of complex concepts and facilitates more effective knowledge acquisition for different sources of information (Wei et al., 2022). Nowadays, multi-modal learning has demonstrated its remarkable ability in various tasks, including scene understanding (Antol et al., 2015; Yang et al., 2022; Li et al., 2022a), and emotion recognition (Tripathi et al., 2018; Chudasama et al., 2022).

However, real-world data are often perturbed, such as attacks, and missing modality, which impacts the performance of multi-modal model (Kumar et al., 2020). As a result, multi-modal robustness, which refers to the model's ability to defend against such perturbations, has received increasing attention in recent studies (Bednarek et al., 2020; Vishwamitra et al., 2021). Unlike data from a single modality, multi-modal data can be perturbed across all modalities. Therefore, for robustness, multi-modal models should have the ability to resist attacks on both individual and multiple modalities. Nevertheless, experiments have suggested that multi-modal models could perform badly when encountering perturbation (Noever & Noever, 2021) or missing modality (Yu et al., 2020; Ma et al., 2022). Based on these observations about the vulnerability of the multi-modal model, how to achieve a robust multi-modal model arises and attaches more focus.

To address this, previous methods improve the training strategies to obtain a robust multi-modal model (Liang et al., 2021; Ding et al., 2021). Specifically, certain studies extend uni-modal robust training strategies, like adversarial training and mixup, to learn a discriminative multi-modal decision boundary (Li et al., 2022b; Maheshwari et al., 2023). Others take steps like cross-modal

---

*Corresponding author

alignment (Tian & Xu, 2021) to enhance the connection among modalities and obtain compact and robust multi-modal representation. However, even if they empirically improve robustness to some degree, they still lack in-depth theoretical analysis to understand the resilience of multi-modal models to perturbations, which is vital for safety-critical applications. More importantly, a universal phenomenon can be observed for these robust training methods that the adversarial attack on specific modalities could be more effective than others. As shown in Figure 1, the $\ell_2$-PGD attack is more effective on modality #$a$ than modality #$v$ for both Joint Training and three widely used robust training methods on the Kinetics Sounds dataset.

To deeply understand multi-modal robustness and elucidate this phenomenon, we first depict the multi-modal decision boundary by integrating uni-modal representation margins. Then, we derive a lower bound for the perturbation radius that the multi-modal model can consistently defend against. We discover that larger uni-modal margins coupled with reasonable integration are crucial for enhanced multi-modal robustness. With the above theoretical results, we investigate the pervasive issue (Wang et al., 2020; Peng et al., 2022) that multi-modal models exhibit a pronounced preference for a particular modality. This preferred modality profoundly influences the model's decision, then resulting in the above robustness phenomenon. On one hand, when a specific preferred modality is substantial enough to rely upon, the model will be reluctant to learn from other modalities, leading to the imbalance problem (Huang et al., 2022; Wu et al., 2022). This imbalance problem hinders the enhancement of the uni-modal margin on the

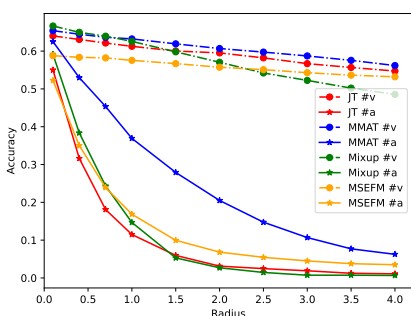

Figure 1: Accuracy of different multi-modal robust training methods compared with Joint Training (JT) baseline under $\ell_2$-PGD attack with a range of radius for modality #$v$ (vision) and #$a$ (audio) respectively on Kinetics Sounds dataset. Results show that all these methods are more vulnerable to attacks on the specific modality #$a$.

other modality, thus limiting the robustness. On the other hand, since the multi-modal model heavily relies on the preferred modality, the corresponding factor used in modality integration becomes larger, which will amplify the variation of the uni-modal margin in decision-making. Hence, in case the preferred modality is vulnerable, the multi-modal model becomes more vulnerable to multi-modal attack. Further, this preference for vulnerable modality makes attacks on the preferred modality significantly more effective than other ones, explaining the observation in Figure 1.

Since the essential components of robustness have an interrelation with each other, directly applying regulation is hard to guarantee higher robustness. To address this, we employ an orthogonal-based framework that formulates an alternative bound, which eliminates the interrelation and explicitly presents the integration. Building upon our theoretical analysis, we introduce a two-step *Certifiable Robust Multi-modal Training* (CRMT) procedure to ensure progressively superior robustness. Initially, we redefine uni-modal margins related to the reliability of the modality in this framework. Then we propose to regulate the unreliable modality by enlarging its margin, which can alleviate the imbalanced problem brought by modality preference. Further, our approach adjusts the integration of modalities considering the improvement of certified bound. These steps not only mitigate the large gap between robustness against attack on each modality but also credibly guarantee higher multi-modal robustness. To validate our method, we conduct extensive experiments to present the advanced robustness against both uni-modal and multi-modal attacks. Our contributions are as follows:

1. We focus on a commonly used multi-modal model, offering invaluable insights into the essential components influencing multi-modal robustness.
2. We present analyses highlighting how multi-modal preference limits the multi-modal robustness and contributes to the vulnerability of multi-modal models towards specific modalities.
3. Drawing from our theoretical findings, we introduce a two-step training procedure, alleviating the limitation brought by modality preference. Our method can effectively enhance both performance and robustness over three real-world multi-modal datasets.

## 2 RELATED WORK

**Multi-modal Robustness Analysis.** Recent studies have highlighted the vulnerability of deep neural networks (DNNs) to attacks and perturbations (Goodfellow et al., 2014; Madry et al., 2017),

raising significant concerns regarding their deployment. With the presence of multiple modalities, the forms of perturbations include uni-modal attacks, multi-modal attacks (Schlarmann & Hein, 2023), and modality missing (Lee et al., 2023). One possible way to enable the models to resist these perturbations is to design robust training strategies to obtain a reliable decision boundary that is distanced from samples (Li et al., 2021). Certain methods, such as multi-modal adversarial training (Li et al., 2022b) and incorporating uni-modal tasks (Ma et al., 2022), have proven pertinent in this context. On the other hand, studies focus on the latent consistency among modalities and propose to enhance modality interaction like alignment (Tsai et al., 2018) to obtain compact and robust representation (Bednarek et al., 2020). Among these empirical works, we identify a universal phenomenon that multi-modal models are commonly vulnerable to a certain modality (Liang et al., 2021), which currently lacks a comprehensive theoretical explanation. This motivates us to theoretically figure out the essential components determining the multi-modal robustness and explain the vulnerable modality phenomenon.

**Multi-modal imbalance problem.** Multi-modal learning is a significant approach for achieving a richer understanding across various tasks (Baltrušaitis et al., 2018; Jiang et al., 2021). However, even with the potential for a more comprehensive latent representation from multi-modal models (Huang et al., 2021), their performance might not always surpass that of the best uni-modal counterparts (Wang et al., 2020; Huang et al., 2022). Due to the inherent differences between modalities, a multi-modal model might prefer or lean towards a modality that is easier to learn, potentially overlooking others (Wu et al., 2022). In response, various strategies are proposed to strengthen the learning of uni-modalities, thereby enhancing the generalization of multi-modal model (Wang et al., 2020; Peng et al., 2022; Fan et al., 2023; Xu et al., 2023b). However, these researches do not elucidate how this modality preference problem impacts robustness against perturbations, which is our main focus.

**Certified robustness.** To describe the robustness of the model, certified robustness is introduced to describe the size of permissible perturbations that a model can consistently defend against. Techniques such as randomized smoothing (Cohen et al., 2019; Rosenfeld et al., 2020; Yang et al., 2021; Salman et al., 2019) and interval bound propagation (Zhang et al., 2018; Lyu et al., 2021; Xu et al., 2020) are widely utilized to relax the model, which assists in determining the certificate bounds. Additionally, the Lipschitz constant can serve as an intuitive metric to illustrate how perturbations influence the decision of models (Weng et al., 2018; Leino et al., 2021). However, these methods are tailored for inputs with only a single modality but fail to focus on the primary concerns with multiple modalities. In our research, we establish certified robustness for multi-modal models and determine the essential components influenced by modality preference that limit the robustness.

## 3 METHOD

### 3.1 PRELIMINARIES

**Multi-modal Framework.** We consider a general $K$-way classification problem with a vectorized input sample $\boldsymbol{x} = (\boldsymbol{x}^{(1)}, \boldsymbol{x}^{(2)}) \in \mathbb{R}^{d_1+d_2}$ consisting of two modalities, and a ground truth label $y \in [K]$. We consider the commonly used joint learning framework in multi-modal learning, where all of the modalities are projected into the shared space for downstream tasks (Huang et al., 2022). Concretely, the uni-modal representations are extracted by encoders $\phi^{(m)}$ and concatenated to form the joint representation, which is then mapped to the output space using a linear classifier, where $W \in \mathbb{R}^{K \times (\dim(\phi^{(1)})+\dim(\phi^{(2)}))}$ and $\boldsymbol{b} \in \mathbb{R}^K$ are the weight matrix and bias respectively, and $\dim(\cdot)$ represents the dimension. The logits output of the multi-modal model can be denoted as $h(\boldsymbol{x}) = W[\phi^{(1)}(\boldsymbol{x}^{(1)}); \phi^{(2)}(\boldsymbol{x}^{(2)})] + \boldsymbol{b} = W^{(1)}\phi^{(1)}(\boldsymbol{x}^{(1)}) + W^{(2)}\phi^{(2)}(\boldsymbol{x}^{(2)}) + \boldsymbol{b}$, where $W^{(m)} \in \mathbb{R}^{K \times \dim(\phi^{(m)})}$ is the part of classifier $W$ related to the $m$-th modality.

**Multi-modal robustness.** To assess the certified robustness of a multi-modal model, we can measure the radius of the smallest perturbation that shifts a sample to the decision boundary. In other words, any perturbation that falls within this radius can be defended against.

**Definition 3.1.** Suppose the multi-modal model $h$ can correctly classify the sample $\boldsymbol{x}$, *i.e.* $\forall k \neq y, h_y(\boldsymbol{x}) > h_k(\boldsymbol{x})$, where $h_k$ denotes the logit score of the $k$-th class. The robustness radius of the multi-modal model towards the sample $\boldsymbol{x}$ can be defined as:

$$P(\boldsymbol{x}) = \min_{\boldsymbol{x}'} \|\boldsymbol{x} - \boldsymbol{x}'\|_2 \qquad s.t. \ \exists j \neq y, h_y(\boldsymbol{x}') = h_j(\boldsymbol{x}'). \tag{1}$$

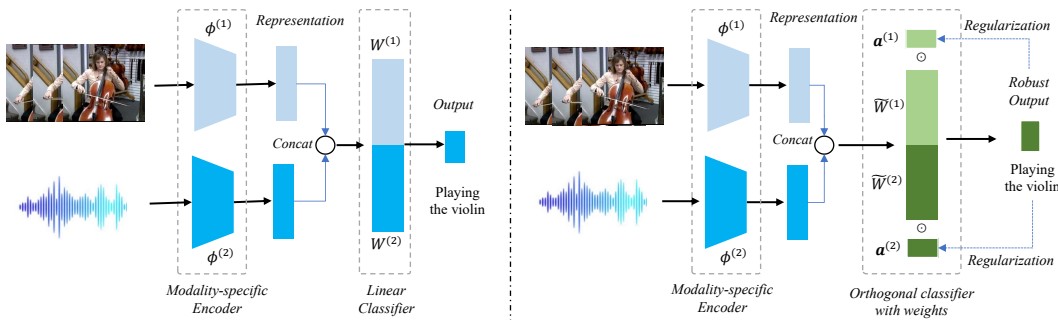

Figure 2: Illustration of traditional multi-modal joint learning framework (Baltrušaitis et al., 2018) (left) and our framework introducing orthogonality into each uni-modal classifier (right). Our framework can be easily applied to explicit regularization to achieve larger certified robustness.

Equation 1 seeks to identify the smallest perturbation, denoted as $\boldsymbol{x} - \boldsymbol{x}'$, that results in reaching the decision boundary between the ground truth $y$ and its nearest class $j$. Hence, any smaller perturbation can always be defended against. Here, we use $\ell_2$-norm to measure the radius, reflecting the overall size of the perturbation (Tsuzuku et al., 2018; Carlini & Wagner, 2017).

## 3.2 CERTIFIED ROBUSTNESS FOR MULTI-MODAL MODEL

In this section, we endeavor to uncover the essential components that impact multi-modal robustness. Referring to Figure 1, there is a notable variation in the effectiveness of perturbations on different modalities. This observation prompts us to distinguish the differences within each uni-modality. Therefore, we introduce the concept of *uni-modal margin*, which quantifies the distance between a uni-modal representation and the uni-modal decision boundary. To elaborate further, $(W_{y\cdot}^{(m)} - W_{k\cdot}^{(m)})\phi(\boldsymbol{x}^{(m)}) = 0$ signifies that sample $\boldsymbol{x}^{(m)}$ is positioned on the decision boundary between classes $y$ and $k$ for $m$-th modality. The uni-modal representation margin can be defined as:

**Definition 3.2.** Given the uni-modal encoder $\phi^{(m)}$ and the classifier $W^{(m)}$, the margin on representation space between ground truth $y$ and other label $k$ is defined as:

$$\zeta_k^{(m)}(\boldsymbol{x}^{(m)}) = \frac{(W_{y\cdot}^{(m)} - W_{k\cdot}^{(m)})\phi^{(m)}(\boldsymbol{x}^{(m)})}{\left\|W_{y\cdot}^{(m)} - W_{k\cdot}^{(m)}\right\|_2}. \tag{2}$$

In this context, a larger margin indicates the uni-modality is more reliable in distinguishing these two classes. Using this margin definition, we can re-examine the constraint conditions in Equation 1, which outline the perturbed sample located on the multi-modal decision boundary where $y$ and the closest class $j$ are tied in the output space:

$$h_y(\boldsymbol{x}') - h_j(\boldsymbol{x}') = c_j^{(1)}\zeta_j^{(1)}(\boldsymbol{x}'^{(1)}) + c_j^{(2)}\zeta_j^{(2)}(\boldsymbol{x}'^{(2)}) + \beta_j = 0. \tag{3}$$

Inspired by Equation 3, we observe that the multi-modal decision boundary can be described as integration of different uni-modal margins of the perturbed sample with factors $c_j^{(m)} = \|W_{y\cdot}^{(m)} - W_{j\cdot}^{(m)}\|_2$ and constant term $\beta_j = b_y - b_j$. The integration factors $c_j^{(m)}$ quantify how variations in the uni-modal margin influence multi-modal decisions. When perturbing a sample $\boldsymbol{x}$ towards the decision boundary, a larger factor $c_j^{(m)}$ indicates altering the margin of the corresponding modality is more effective. Meanwhile, when the sample is perturbed, how the uni-modal margin varies also depends on the size of the perturbation. Thus, we propose to introduce the Lipschitz constant $\tau_j^{(m)}$ for the uni-modal margin. This Lipschitz constant is the smallest constant to limit the local variation range of a certain function (Finlay et al., 2018), which is given by:

$$|\zeta_j^{(m)}(\boldsymbol{x}^{(m)}) - \zeta_j^{(m)}(\boldsymbol{x}'^{(m)})| \leq \tau_j^{(m)}\left\|\boldsymbol{x}^{(m)} - \boldsymbol{x}'^{(m)}\right\|_2. \tag{4}$$

Then, we can provide the lower bound for the perturbation radius for multi-modal robustness.

**Theorem 3.3.** *Given an input $\boldsymbol{x}$ with ground-truth label $y \in [K]$ and the closest label $j \neq y$, $\zeta_j^{(m)}(\boldsymbol{x}^{(m)})$ as the representation margin for $m$-th modality with Lipschitz constraint $\tau_j^{(m)}$, and the*

integration factor $c_j^{(m)}$. *Define $\boldsymbol{x}'$ as the perturbed sample, and $\boldsymbol{x} - \boldsymbol{x}'$ as the perturbation. The lower bound for the perturbation radius can be described as:*

$$P(\boldsymbol{x}) = \min_{\boldsymbol{x}'} \|\boldsymbol{x} - \boldsymbol{x}'\|_2 \geq \frac{c_j^{(1)}\zeta_j^{(1)}(\boldsymbol{x}^{(1)}) + c_j^{(2)}\zeta_j^{(2)}(\boldsymbol{x}^{(2)}) + \beta_j}{\sqrt{(c_j^{(1)}\tau_j^{(1)})^2 + (c_j^{(2)}\tau_j^{(2)})^2}} \tag{5}$$

$$where \quad j \neq y \quad s.t. \quad c_j^{(1)}\zeta_j^{(1)}(\boldsymbol{x}'^{(1)}) + c_j^{(2)}\zeta_j^{(2)}(\boldsymbol{x}'^{(2)}) + \beta_j = 0.$$

The detailed proof can be found in the Appendix 7.1. Based on the certified bound above, we deduce that the multi-modal robustness depends on three primary factors: uni-modal representation margins $\zeta_j^{(m)}$, integration $c_j^{(m)}$, and bias difference $\beta_j$. Firstly, robustness increases proportionally with the enhancement of the uni-modal representation margin $\zeta_j^{(m)}$. Secondly, the integration $c_j^{(m)}$ is related to both uni-modal margins and the uni-modal Lipschitz constant, the expected integration for robustness requires considering both. Thus, a reasonable choice of integration factor can benefit higher robustness. Thirdly, when the sample is perturbed, the bias difference term $\beta_j$ stays invariant, since it depends on class $y$ and $j$ rather than the specific sample. Based on this fact, the bias difference is not considered in our analysis about model robustness. In a nutshell, we recognize that uni-modal margins and the integration of modalities are two essential components. In the following section, we will analyze how these essential components vary and influence the robustness of multi-modal models, especially under modality preference.

### 3.3 ANALYSIS ABOUT MODALITY PREFERENCE FOR ROBUSTNESS

Since modalities have different amounts of information, some specific modalities might hold more significance than others in decision-making processes (Gat et al., 2021). As a result, it is widely recognized that multi-modal models tend to show a preference for and predominantly rely on specific modality (Huang et al., 2022; Wu et al., 2022). However, this preference or over-reliance on the specific modality poses challenges for achieving a robust multi-modal model. We delve deeper into how such preferences impact the two essential components of the certified bound in Equation 5.

**Uni-modal representation margin.** As shown in Equation 5, a larger uni-modal margin $\zeta_j^{(m)}(\boldsymbol{x}^{(m)})$ determines higher certified robustness. However, when the learned information in the preferred modality is sufficiently reliable, the multi-modal model is reluctant to learn more information from other modalities (Wu et al., 2022; Huang et al., 2022). Thus, the modality preference leads to an imbalance problem hindering the development of uni-modal representations, resulting in a narrower representation margin and ultimately constraining the certified robustness of the multi-modal model.

**Integration of modalities.** According to modality preference, the decision of the multi-modal model highly depends on a specific modality. Thus, considering the integration of modalities, the preferred modality contributes more and is allocated a larger integration factor $c_j^{(m)}$, which could amplify the variation of the uni-modal margin in multi-modal decision-making. Since the preference is only determined by whether the modality with ideal discriminative ability, the multi-modal model could prefer a vulnerable modality, which has a larger $\tau_j^{(m)}$. Thus, the perturbation for this preferred but vulnerable modality leads to larger variations in multi-modal margins, which is further amplified in decision-making. Motivated by this phenomenon, we define $\eta^{(m)} = c_j^{(m)}\tau_j^{(m)}$ as the vulnerability indicator of modality $m$. When the multi-modal model exhibits a preference for a vulnerable modality 1, there is a significant imbalance in this indicator, with $\eta^{(1)} \gg \eta^{(2)}$. Consequently, we observe:

$$P(\boldsymbol{x}) \geq \frac{c_j^{(1)}\zeta_j^{(1)}(\boldsymbol{x}^{(1)}) + c_j^{(2)}\zeta_j^{(2)}(\boldsymbol{x}^{(2)}) + \beta_j}{\sqrt{(\eta^{(1)})^2 + (\eta^{(2)})^2}} \approx \frac{c_j^{(1)}\zeta_j^{(1)}(\boldsymbol{x}^{(1)}) + c_j^{(2)}\zeta_j^{(2)}(\boldsymbol{x}^{(2)}) + \beta_j}{\eta^{(1)}}. \tag{6}$$

That is to say, the modality preference on vulnerable modality leaves an unreasonable imbalance on $\eta^{(m)}$, thus the multi-modal robustness is highly dependent on the modality with a larger vulnerability indicator. In this way, distinctly attacking the vulnerable modalities alone is enough to obscure the model. Here we further provide the multi-modal robustness under uni-modal attack case:

**Proposition 3.4.** *Following the setting in Theorem 3.3, w.l.o.g. considering the perturbation on* 1-*th modality* $\boldsymbol{x}'^{(1)}$, *the lower bound for the uni-modal perturbation radius can be described as:*

$$\min_{\boldsymbol{x}'^{(1)}} \left\| \boldsymbol{x}^{(1)} - \boldsymbol{x}'^{(1)} \right\|_2 \geq \frac{c_j^{(1)} \zeta_j^{(1)}(\boldsymbol{x}^{(1)}) + c_j^{(2)} \zeta_j^{(2)}(\boldsymbol{x}^{(2)}) + \beta_j}{c_j^{(1)} \tau_j^{(1)}} \tag{7}$$

$$where \quad j \neq y \quad s.t. \quad c_j^{(1)} \zeta_j^{(1)}(\boldsymbol{x}'^{(1)}) + c_j^{(2)} \zeta_j^{(2)}(\boldsymbol{x}^{(2)}) + \beta_j = 0.$$

The lower bounds of different uni-modal perturbations share identical numerators but differ in denominators, the vulnerability indicator $\eta^{(m)}$. The larger indicator on preferred modality reduces the lower bound for perturbations on this modality, thus making attacks on this preferred modality more effective, explaining the observation in Figure 1.

## 3.4 CERTIFIABLE ROBUST MULTI-MODAL TRAINING

Based on the above analysis, we target to improve the uni-modal representation margin $\zeta_j^{(m)}$ and adjust the integration of modalities $c_j^{(m)}$ as regulation for higher certified robustness. However, these regulations are intricately linked with the linear classifier $W$, potentially leading to conflicts in optimization objectives. As a result, stably enhancing the certified robustness presents challenges. Additionally, determining the Lipschitz constant for the margin is computationally demanding. To address these challenges, we propose to adopt orthogonality (Huang et al., 2018) within uni-modal linear classifiers as our framework. Detailedly, we ensure that in each modality, the class-specific vectors $\tilde{W}_{k\cdot}^{(m)}, k \in [K]$ are unit and orthogonal. Since the valuable information between modalities is different, we apply the weight $\boldsymbol{a}^{(m)} \in \mathbb{R}^K$ to lead the model focusing on more reliable modalities. Integrating these weights can enhance the model's ability to effectively utilize the valuable information from each modality. Therefore, learning of uni-modal representations and integration of modalities can be decoupled. The score corresponding to the $k$-th class can be expressed as:

$$\tilde{h}_k(\boldsymbol{x}) = a_k^{(1)} \tilde{W}_{k\cdot}^{(1)} \phi^{(1)}(\boldsymbol{x}^{(1)}) + a_k^{(2)} \tilde{W}_{k\cdot}^{(2)} \phi^{(2)}(\boldsymbol{x}^{(2)}) + \tilde{b}_k, \tag{8}$$

where $\tilde{W}^{(m)} \in \mathbb{R}^{K \times \dim(\phi^{(m)})}$ is the matrix with orthogonal rows, satisfying $\tilde{W}^{(m)} (\tilde{W}^{(m)})^T = I_K$. We can use $\tilde{W}_{k\cdot}^{(m)} \Phi(\boldsymbol{x}^{(m)})$ as the uni-modal score, which represents whether the uni-modal representation is well learned toward the $k$-th class. With this framework in place, we are able to express the new Lipschitz constant $\tilde{\tau}_k^{(m)}$ for the uni-modal score on class $k$ in the $m$-th modality as:

$$|\tilde{W}_{k\cdot}^{(m)} \phi^{(m)}(\boldsymbol{x}^{(m)}) - \tilde{W}_{k\cdot}^{(m)} \phi^{(m)}(\boldsymbol{x}'^{(m)})| \leq \tilde{\tau}_k^{(m)} \left\| \boldsymbol{x}^{(m)} - \boldsymbol{x}'^{(m)} \right\|_2. \tag{9}$$

With these definitions, we can derive the certified bound for the multi-modal perturbation radius, which is distinctly tailored to the framework employing orthogonal classifiers:

**Theorem 3.5.** *Given an input $\boldsymbol{x}$ with ground-truth label $y \in [K]$ and the closest label $j \neq y$, the orthogonal classifier $\tilde{W}^{(m)}$, the modality-specific weight $\boldsymbol{a}^{(m)}$, the Lipschitz constant $\tilde{\tau}_j^{(m)}$, and the difference of the bias $\tilde{\beta}_j = \tilde{b}_y - \tilde{b}_j$. The lower bound for the perturbation radius with the orthogonal-based framework can be described as:*

$$P(\boldsymbol{x}) \geq \frac{\sum_{m=1}^2 \left( a_y^{(m)} \tilde{W}_{y\cdot}^{(m)} \phi^{(m)}(\boldsymbol{x}^{(m)}) - a_j^{(m)} \tilde{W}_{j\cdot}^{(m)} \phi^{(m)}(\boldsymbol{x}^{(m)}) \right) + \tilde{\beta}_j}{\sqrt{\sum_{m=1}^2 \left( a_y^{(m)} \tilde{\tau}_y^{(m)} + a_j^{(m)} \tilde{\tau}_j^{(m)} \right)^2}} \tag{10}$$

$$where \quad j \neq y, \quad s.t. \quad \tilde{h}_y(\boldsymbol{x}') = \tilde{h}_j(\boldsymbol{x}').$$

See Appendix 7.2 for proof. With this lower bound, we can discuss designing a regulation method to explicitly enhance the certified robustness. Firstly, as analyzed in Section 3.3, the imbalance problem brought by modality preference impacts the unreliable modal representation margin. To explicitly regulate the uni-modal encoder and classifier independent from integration, we redefine the margin as the difference in uni-modal score between ground truth and the other label, which can also reflect the reliability of uni-modality. Then we propose to enlarge the relatively small margin through regularization, which is expressed as follows:

$$\max_{\tilde{W}^{(m)}, \phi^{(m)}} \min_{m; k \neq y} \quad \tilde{W}_{y\cdot}^{(m)} \phi^{(m)}(\boldsymbol{x}^{(m)}) - \tilde{W}_{k\cdot}^{(m)} \phi^{(m)}(\boldsymbol{x}^{(m)}). \tag{11}$$

Table 1: Experimental results of adversarial accuracy results. Bold and underlined values indicate the top and the runner-up results. Combined with JT, MMAT, and Mixup, our proposed CRMT-based methods can improve both performance and robustness.

| Attack Datasets | w\o | | | FGM | | | $\ell_2$ PGD | | |
|---|---|---|---|---|---|---|---|---|---|
| | KS | UCF101 | VGGS | KS | UCF101 | VGGS | KS | UCF101 | VGGS |
| JT | 0.643 | 0.742 | 0.496 | 0.288 | 0.506 | 0.157 | 0.268 | 0.431 | 0.056 |
| GB | 0.704 | 0.784 | 0.529 | 0.371 | 0.432 | 0.279 | 0.344 | 0.205 | 0.174 |
| OGM | 0.651 | 0.743 | 0.498 | 0.292 | 0.492 | 0.190 | 0.270 | 0.377 | 0.069 |
| PMR | 0.689 | 0.742 | 0.503 | 0.350 | 0.455 | 0.166 | 0.331 | 0.292 | 0.059 |
| MSEFM | 0.627 | 0.721 | 0.492 | 0.390 | 0.483 | 0.187 | 0.376 | 0.243 | 0.115 |
| MMAT | 0.656 | 0.728 | 0.509 | 0.514 | 0.609 | 0.413 | 0.507 | 0.598 | 0.409 |
| Mixup | 0.669 | 0.717 | 0.507 | 0.347 | 0.413 | 0.278 | 0.327 | 0.192 | 0.135 |
| CMRT | 0.758 | **0.789** | 0.526 | 0.491 | 0.515 | 0.248 | 0.468 | 0.433 | 0.124 |
| CMRT-AT | **0.762** | 0.759 | **0.538** | **0.608** | **0.614** | **0.422** | **0.602** | **0.602** | **0.414** |
| CMRT-Mix | 0.744 | 0.769 | 0.537 | 0.514 | 0.430 | 0.328 | 0.491 | 0.191 | 0.184 |

Based on Equation 11, we propose a regularization term, refining the maximum using the LogSumExp function to facilitate better optimization:

$$L_1 = \frac{1}{N} \sum_{i=1}^{N} \log \left( \sum_{m=1}^{2} \frac{\sum_{k \neq y} \exp(\tilde{W}_{k \cdot}^{(m)} \phi^{(m)}(\boldsymbol{x}_i^{(m)}))}{\exp(\tilde{W}_{y \cdot}^{(m)} \phi^{(m)}(\boldsymbol{x}_i^{(m)}))} \right), \tag{12}$$

which is detailed in the Appendix 7.3. Furthermore, we can adjust the integration of different modalities by enlarging the lower bound in Equation 10. Subsequently, we propose a two-step training procedure called *Certifiable Robust Multi-modal Training* (CRMT), which can credibly obtain a robust multi-modal model. The training procedure of CRMT is as follows:

Step 1: optimize with cross-entropy loss and margin regularization with term $\rho$:
$\min_{\boldsymbol{a}^{(m)}, \tilde{W}^{(m)}, \phi^{(m)}} \rho L_1 + \frac{1}{N} \sum_{i=1}^{N} CE(h(\boldsymbol{x}_i), y_i)$, where $CE$ is the cross-entropy loss function.

Step 2: fix $\tilde{W}^{(m)}, \phi^{(m)}$, update $\boldsymbol{a}^{(m)}$ to approach higher certified robustness:
$\min_{\boldsymbol{a}^{(m)}} \quad L_2 = -\frac{1}{N} \sum_{i=1}^{N} r(\boldsymbol{x}_i)$, where $r(\boldsymbol{x})$ is the lower bound in Equation 10.

As shown in Appendix 8.3, we further demonstrate that only one iteration of our method can achieve considerable robustness, without a huge consumption of training cost.

# 4 EXPERIMENTS

## 4.1 SETUPS

**Dataset.** We evaluate our method on different datasets including Kinetics-Sounds (Audio + Vision) (Arandjelovic & Zisserman, 2017), UCF101 (Optical flow + RGB) (Soomro et al., 2012), and VGGSound (Audio + Vision) (Chen et al., 2020). We use the backbone ResNet18 (He et al., 2016) as the encoder for each uni-modality. Details about these datasets are presented in Appendix 8.1.

**Multi-modal models.** In this study, the comparison methods are selected to improve multi-modal learning and be suitable for the multi-modal joint training strategy. Comparison methods can be divided into two distinct groups: Firstly, methods address the imbalance problem caused by modality preference: *Gradient Blending* (GB) (Wang et al., 2020), *On-the-fly Gradient Modulation with generalization enhancement* (OGM) (Peng et al., 2022), *Prototypical Modal Rebalance* (PMR) (Fan et al., 2023). Secondly, methods aim at improving multi-modal robustness: *Multi-Modal Adversarial Training* (MMAT) (Li et al., 2022b), *Multi-modal mixup* (Mixup) (Madry et al., 2017; Li et al., 2022b), *MinSim+ExFMem* (MSEFM) (Tian & Xu, 2021). Our method can be extended to different training strategies, denoted as *Certifiable Robust Multi-modal Training with Joint Training* (CRMT-JT), CRMT *with Adversarial Training* (CRMT-AT), and CRMT *with Mixup* (CRMT-Mix).

**Attack methods.** Following previous work (Tsuzuku et al., 2018; Singla et al., 2021), we select *Fast Gradient Method* (FGM) (Goodfellow et al., 2014) and $\ell_2$ *Projected Gradient Descent* ($\ell_2$-PGD) (Madry et al., 2017) as two attack methods with attack size $\epsilon = 0.5$, which are widely used

Table 2: Performance against distinct uni-modal attack methods on KS dataset.

| Method | Attack | w\o Clean | FGM #v | FGM #a | $\ell_2$-PGD #v | $\ell_2$-PGD #a | Missing modality #v | Missing modality #a |
|---|---|---|---|---|---|---|---|---|
| Baseline | JT | 0.643 | 0.616 | 0.143 | 0.616 | 0.110 | 0.480 | 0.230 |
| Imbalance | GB | 0.704 | 0.658 | 0.195 | 0.656 | 0.157 | 0.483 | 0.430 |
| | OGM | 0.651 | 0.624 | 0.147 | 0.624 | 0.116 | 0.463 | 0.207 |
| | PMR | 0.689 | 0.649 | 0.181 | 0.649 | 0.152 | 0.484 | 0.315 |
| Robustness | MSEFM | 0.627 | 0.577 | 0.320 | 0.576 | 0.314 | 0.466 | 0.239 |
| | MMAT | 0.656 | 0.633 | 0.390 | 0.632 | 0.369 | 0.482 | 0.262 |
| | Mixup | 0.669 | 0.628 | 0.191 | 0.626 | 0.147 | 0.450 | 0.256 |
| Ours | CRMT-JT | 0.758 | 0.685 | 0.384 | 0.682 | 0.327 | 0.560 | 0.591 |
| | CRMT-AT | **0.762** | 0.703 | **0.488** | 0.698 | **0.464** | 0.547 | **0.626** |
| | CRMT-Mix | 0.744 | **0.706** | 0.377 | **0.704** | 0.320 | **0.572** | 0.566 |

for verifying multi-modal robustness with $\ell_2$-norm. For uni-modal attack, we also introduce attacks FGM and $\ell_2$-PGD with $\epsilon = 1.0$, and missing on uni-modality.

## 4.2 ROBUSTNESS VALIDATION ON MULTI-MODAL AND UNI-MODAL ATTACK

**Robustness against multi-modal attacks.** We validate the robustness of our method under multi-modal attacks. Based on the experimental results presented in Table 1, we have identified four key observations that warrant attention. Firstly, while imbalance methods effectively enhance performance on clean samples, robustness methods can demonstrate more notable defense capabilities against various attacks. Secondly, the imbalance method GB can be superior to the robustness method MSEFM under some situations. That is because GB introduces a multi-task method to enhance the learning of uni-modality and improve the uni-modal margin, thus it can enhance the robustness methods according to our analysis. Thirdly, our proposed CRMT-based methods surpass the performance of the compared methods across these datasets in most cases. This superiority stems from our approach to addressing the imbalance problem through improving uni-modal representation margins, and the certificate adjustment of modality-specific weights for heightened certified robustness. Fourthly, the improvement of results for CRMT-AT and CRMT-Mix suggests that our training procedure can serve as a valuable component applicable to other robust training methods.

**Robustness against distinct uni-modal attack.** To substantiate the efficacy of our CRMT methods, we conducted additional experiments on the KS dataset, involving a broader range of different uni-modal attacks. As demonstrated in Table 2, the absence of modality #a leads to a larger performance decline than #v, since it is more preferred by multi-modal models. As shown in Table 2, previous multi-modal methods demonstrated poor performance when subjected to attacks on the preferred modality #a, aligning with our analysis. In contrast, our CRMT-based methods consider addressing the imbalance problem introduced by modality preference and adjusting integration, thus making the model more robust against uni-modal attacks. Furthermore, when encountering different uni-modal attacks, our CRMT-based approach shows superior performance and consistently ensures higher robustness in various scenarios. This strongly demonstrates the effectiveness and versatility of our proposed method in enhancing the robustness of multi-modal models.

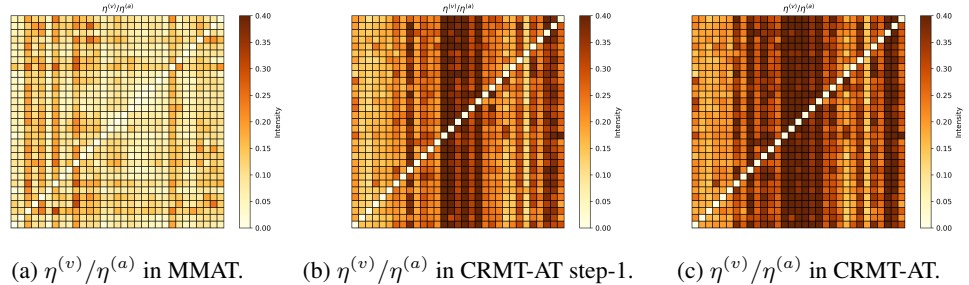

(a) $\eta^{(v)}/\eta^{(a)}$ in MMAT.    (b) $\eta^{(v)}/\eta^{(a)}$ in CRMT-AT step-1.    (c) $\eta^{(v)}/\eta^{(a)}$ in CRMT-AT.

Figure 3: Evaluation of the ratio of vulnerability indicators between modality #v and #a (preferred). We illustrate the ratio in MMAT, CRMT-AT, and CRMT-AT with only the first training procedure.

Table 3: Extension results of adversarial accuracy with transformer-based method on VGGS dataset.

| Attack | Clean | Uni-modal attack | | | | Multi-modal attack | |
|---|---|---|---|---|---|---|---|
| Method | w\o | FGM #v | FGM #a | $\ell_2$-PGD #v | $\ell_2$-PGD #a | FGM | $\ell_2$-PGD |
| MMT | 0.465 | 0.433 | 0.259 | 0.431 | 0.228 | 0.331 | 0.326 |
| CRMT-MMT | 0.471 | 0.440 | 0.276 | 0.428 | 0.237 | 0.344 | 0.340 |

**Imbalance on uni-modal vulnerability indicators.** We verify the conclusion drawn in Section 3.3 that the imbalance problem introduced by modality preference leads to the vulnerability of attack on specific modality, as illustrated in Figure 1. To achieve this, we compare MMAT with our CRMT-AT approach. As depicted in Equation 7, $\eta^{(m)}$ is utilized as a measurement to assess the robustness of multi-modal models against uni-modal perturbations, and we employ the ratio of uni-modal vulnerability indicators $\eta^{(v)}/\eta^{(a)}$ (see Figure 3). It can be demonstrated that our approach diligently works to reduce the imbalance in the indicator $\eta$, effectively darkening the heated map. Furthermore, according to the robustness accuracy, our approach significantly reduces the disparity in attack performance between the two modalities (see Figure 4). This evidence serves to illustrate the efficacy of our method in mitigating the imbalance problem and ultimately enhancing multi-modal robustness.

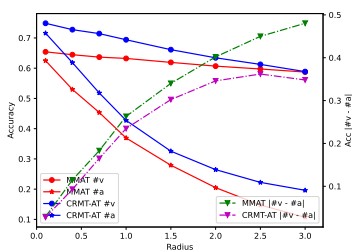

Figure 4: This figure presents the robustness accuracy against uni-modal attacks with different sizes, where the dotted line signifies the difference in robustness accuracy between two uni-modalities.

### 4.3 VALIDATION FOR EFFECTIVENESS AND SCALABILITY

**Ablation studies.** To delve deeper into the efficacy of our training procedure, we conduct an ablation analysis, aiming to elucidate the contribution of each component of the proposed training procedure toward the overall result. When compared with the baseline Joint Training (JT), our results include the orthogonal-based framework and our proposed training procedure. We report our findings related to solely using the orthogonal framework (OJT), CRMT focusing exclusively on each step (CRMT-step-1 and CRMT-step-2), and our full proposal, CRMT-JT, which incorporates both steps. As depicted in Figure 5, it is evident that the orthogonal-based framework does not contribute to improved robustness. Meanwhile, both step-1 and step-2 of the training procedure can enhance robustness, particularly in the face of larger-size attacks. Furthermore, the results highlight that each step consistently enhances multi-modal robustness, making CRMT-JT superior in multi-modal robustness.

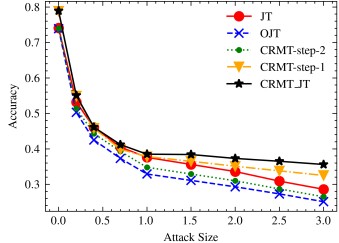

Figure 5: Ablation studies of our methods on the UCF101 dataset, revealing the effect of each part we introduced.

**Extension studies to transformer.** We further provide experiment results of our methods extended to the Multi-Modal Transformer-based framework with hierarchical attention (Xu et al., 2023a) (MMT) on the VGGS dataset. Both visual and audio class tokens are concatenated and linearly projected into the output space. We mainly evaluate the robustness under the Transformer model accurately, hence we train from scratch instead of using the pre-trained model. To validate our method, we only introduce the CRMT procedure step-1 on this transformer architecture as a comparison. That is because both audio and visual input influence each token, which is beyond our assumption. Based on the results in Table 3, our method can also improve the robustness of transformer-based architecture in most cases, which proves the broad prospects of our approach.

## 5 CONCLUSION

In this study, we present the essential components for multi-modal robustness and delve into the limitations imposed by modality preference. Additionally, we explore why multi-modal models exhibit vulnerabilities to attack specific modalities. Further, we introduce the Certifiable Robust Multi-modal Training procedure, a novel approach explicitly designed to boost the certified robustness.

## 6 ACKNOWLEDGEMENT

This research was supported by National Natural Science Foundation of China (NO.62106272), the Young Elite Scientists Sponsorship Program by CAST (2021QNRC001), and Public Computing Cloud, Renmin University of China.

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
