# OpenReview forum: "Quantifying and Enhancing Multi-modal Robustness with Modality Preference"
_ICLR.cc/2024/Conference — ICLR 2024 poster_

### Official Review · Reviewer_hY8P · 2023-10-31

**Soundness:** 3 good
**Presentation:** 3 good
**Contribution:** 3 good
**Rating:** 6
**Confidence:** 4

**Summary:**

This manuscript concerns the robust multi-modal representation learning, which are positioned well away from the discriminative multi-modal decision boundary. To address this issue, they theoretically discover that larger uni-modal representation margins and more reliable integration for modalities are essential components for achieving higher robustness. Inspired by our theoretical finding, we introduce a training procedure called Certifiable Robust Multi-modal Training (CRMT), which can alleviate this influence from modality preference and explicitly regulate essential components to significantly improve robustness in a certifiable manner. Experiments validate the effectiveness.

**Strengths:**

1.	The multi-modal robustness learning is meaningful and challenging. The paper is well-written and the proposed method is easy to understand.
2.	The authors theoretically discover that larger uni-modal representation margins and more reliable integration for modalities are essential components for achieving higher robustness.
3.	Experiments on various datasets validate the proposed method.

**Weaknesses:**

The manuscript claims that they focus on the commonly used joint multi-modal framework, more multi-modal fusion method, and different multi-modal backbones should be compared. For example, the early fusion, and hybrid fusion strategy. On the other hand, different modalities can employ various backbones, the reviewer is curious about the influence of different backbones, and more ablation studies are expected.

In related work and comparison methods, more state-of-the-art multi-modal robustness approaches should be introduced and compared.

How can this setup be extended to three modalities? More explanations and experiments are needed.

**Questions:**

refer to the weakness

---

> ### Author Response · Authors · 2023-11-19
> **Response by authors**
>
> Thank you for your valuable feedback and insightful comments! We respond to some concerns below:
>
> 1. **Different multi-modal fusion backbones**
>
> Thank you for your advice. In our study, we focus on intermediate fusion by employing the widely used MultiModal Transfer Module (MMTM) [1] and conducting experiments on the Kinetic-Sounds dataset. The results from these experiments indicate that our method is well-suited for more multi-modal fusion strategies.
>
>
> |   Kinetic-Sounds   |  Clean  | Missing #v | Missing #a |   FGM   |  PGD-$\ell_2$ |
> |:--------:|:-------:|:---------:|:---------:|:-------:|:-------:|
> | MMTM | 0.6693  |  0.4542   |  0.2028   | 0.3438  | 0.3205  |
> |   CRMT-MMTM (ours)   | 0.6737  |  0.5211   |  0.3169   | 0.3445  | 0.3358  |
>
>
> [1] H. R. Vaezi Joze, A. Shaban, M. L. Iuzzolino, and K. Koishida, “Mmtm: Multimodal transfer module for cnn fusion,” in *Conference on Computer Vision and Pattern Recognition*(CVPR), 2020.
>
> 2. **Different modalities can employ various backbones**
>
> Thank you for pointing it out. We carry out experiments with different backbones on the Kinetic-Sounds dataset, specifically by replacing the audio backbone with a Transformer. The results, detailed in the table below, empirically validate our approach's effectiveness across different backbone architectures.
>
> |  ResNet18 (V) + Transformer (A)  |  Clean  | Missing #v | Missing #a |   FGM   |  PGD-$\ell_2$ |
> |:--------:|:-------:|:---------:|:---------:|:-------:|:-------:|
> | JT | 0.5538  |  0.3387   |  0.2703   | 0.2456  | 0.2100  |
> |  CRMT-JT(ours) | 0.5807  |  0.3721   |  0.3539   | 0.3067  | 0.2711  |
>
>
> 3. **State-of-the-art multi-modal robustness approaches**
>
> Thank you for your suggestion. We compare our method with two recently proposed methods; Robust Multi-Task Learning (RMTL) [1] and Uni-Modal Ensemble with Missing Modality Adaptation (UME-MMA) [2]. For RMTL, we apply the idea of multi-task learning to obtain a robust multi-modal model, which includes full-modal tasks, and modality-specific tasks. We apply these attacks on the Kinetic-Sounds dataset and report the result to show the effectiveness of our method.
>
> |  Method |  Clean  | Missing #v | Missing #a |   FGM   |  PGD-$\ell_2$ |
> |:-------:|:-------:|:----------:|:----------:|:-------:|:-------:|
> |   RMTL  | 0.6672  |   0.5015   |   0.2994   | 0.3641  | 0.3445  |
> | UME-MMA | 0.6999  |   0.5334   |   0.4666   | 0.3394  | 0.3125  |
> | CRMT-JT (ours) | 0.7580  |   0.5596   |   0.5908   | 0.4906  | 0.4680  |
>
> [1] M. Ma, J. Ren, L. Zhao, D. Testuggine, and X. Peng, “Are multi-modal transformers robust to missing modality?” in *Proceedings of the IEEE/CVF Conference on Computer Vision and Pattern Recognition*, 2022, pp. 18 177–18 186.
>
> [2] S. Li, C. Du, Y. Zhao, Y. Huang, and H. Zhao, “What makes for robust multi-modal models in the face of missing modalities?” *arXiv preprint arXiv:2310.06383*, 2023.
>
> 4. Extended to **three modalities**? More **explanations and experiments** are needed.
>
> Thank you for your suggestions. In this study, our analysis discusses the universal situation of two modalities, and can also be extended to the scenario with more than two modalities. To consider more modalities, the key is to introduce the margin of these modalities’s representation. Suppose we have $l$ different modality, the input of the $m$-th modality is $\boldsymbol{x}^{(m)}$, define the representation margin $\zeta_{j}^{(m)}(\boldsymbol{x}^{(m)})$, and the corresponding Lipschitz constant $\tau_j^{(m)}$. Thus, our bound can be extended to the following formulation.
>
> \begin{equation}
> \begin{aligned}
>  \min_{\boldsymbol{x}'} \left\| \left\|\boldsymbol{x} - \boldsymbol{x}'\right\| \right\|_2
>  \geq \frac{ \sum _{m=1}^l c_j^{(m)} \zeta_j^{(m)}(\boldsymbol{x}^{(m)}) +  \beta_j}{\sqrt{ \sum _{m=1}^l (c_j^{(m)} \tau_j^{(m)})^2  }} \\\\
>   where   \quad j \neq y \quad s.t. \quad   \sum _{m=1}^l c_j^{(m)} \zeta_j^{(m)}(\boldsymbol{x}'^{(m)}) +  \beta_j = 0.
> \end{aligned}
> \end{equation}
>
> Thus, our approach can analyze the robustness of more than two modalities.
> We conducted experiments on the UCF101 dataset using three modalities: RGB, Optical Flow, and RGB Frame Difference. These experiments were performed both from scratch and with an ImageNet-pretrained ResNet18. The outcomes demonstrate the effectiveness of our method in enhancing multi-modal robustness.
>
> |   Three modality   | JT (Scratch) | CRMT-JT (Scratch, ours) | JT (Pretrained) | CRMT-JT (Pretrained, ours) |
> |:----------:|:--------------------------:|:-------------------------------:|:---------------:|:--------------------:|
> |    Clean   |           0.4490           |             0.5640              |      0.8312     |        0.8506        |
> |     FGM    |           0.4005           |             0.4567              |      0.2471     |        0.4138        |
> |   PGD_$\ell_2$   |           0.3963           |             0.4312              |      0.0783     |        0.2623        |

---

### Official Review · Reviewer_BCRJ · 2023-10-31

**Soundness:** 3 good
**Presentation:** 3 good
**Contribution:** 3 good
**Rating:** 6
**Confidence:** 4

**Summary:**

This work employs an orthogonal-based framework that formulates an alternative bound, eliminating the interrelation and explicitly presenting integration.  Building on the theoretical analysis, they introduce a two-step procedure called Certifiable Robust Multi-modal Training (CRMT) to progressively enhance robustness.

**Strengths:**

(1) Following a more comprehensive analysis, the researchers furnish compelling evidence that demonstrates the constraining effect of multi-modal preference on the robustness of multi-modal systems, which contributes to the vulnerability of multi-modal models to specific modalities.

(2) Building upon their theoretical insights, they present a two-step training protocol designed to alleviate the limitations stemming from modality preference. The suggested approach significantly enhances both the performance and robustness of multi-modal models across different real-world multi-modal datasets.

**Weaknesses:**

(1) Since FGM and PGD are the two white-box attacks chosen in the adversarial robustness experiments, why not consider the stronger white-box Auto Attack?  It is suggested to add the experiment results about Auto Attack in Section4.

(2) "Robustness against multi-modal attacks" mentioned In Section 4.2, since multi-modal attacks are considered, the experimental results in Table 1 only consider single-mode attacks (#a,#v). Is the method proposed in this paper effective when co-attacks (both modality attacks) are existing?  In [1,2], more effective multi-modal attack methods are proposed than uni-modal(such as #a and #v) attack. Can the proposed method effectively resist these multi-modal attack methods? It is suggested that the relevant experiments should be added to Section 4.2, otherwise the conclusion of "Robustness against multi-modal attacks" is somewhat not convincing.


[1]	Zhang J, Yi Q, Sang J. Towards adversarial attack on vision-language pre-training models[C]//Proceedings of the 30th ACM International Conference on Multimedia. 2022: 5005-5013.
[2]	Lu D, Wang Z, Wang T, et al. Set-level Guidance Attack: Boosting Adversarial Transferability of Vision-Language Pre-training Models[J]. arXiv preprint arXiv:2307.14061, 2023.

**Questions:**

No

---

> ### Author Response · Authors · 2023-11-19
> **Response by authors**
>
> Thank you for your valuable feedback and insightful comments! We respond to some concerns below:
>
> 1. Consider the stronger **white-box Auto Attack**? It is suggested to add the experiment results about Auto Attack in Section 4.
>
> Thank you for your advice. In our research, we employed the Fast Gradient Method (FGM) and Projected Gradient Descent with $\ell_2$-norm (PGD-$\ell_2$) to assess model robustness. Due to their scalability and effectiveness, these methods can be readily adapted for multi-modal settings, enabling the generation of both uni-modal and multi-modal attacks. However, the Auto Attack, a complex attack method that combines four distinct attacks, is primarily designed for uni-modal models with a single input and output, making its application to multi-modal settings challenging. Meanwhile, there has been a lack of targeted research on conducting these effective attacks with multiple inputs in the past. We have made some attempts during response, but we still cannot well implement Auto Attack for multiple inputs. Meanwhile, we want to explain that, the conducted experiments in the paper cover various attacks (e.g., multi-modal co-attack, missing condition and uni-modal attack), which are mostly considered by the previous work for multi-modal robustness, and the results can be used to validate the method effectiveness. Even so, we will also conduct research on how to build complex attack like Auto Attack, but in the setting of multiple inputs, in future work. Thanks again for such a valuable suggestion!
>
> 2. **Table 1 only consider single-mode attacks (#a,#v)**. Is the method proposed in this paper effective when **co-attacks (both modality attacks) exist**?
>
> Thank you for your comment. In fact, **the adversarial results in Table 1 is actually the multi-modal attack, where both of the modalities are perturbed**. This multi-modal attack will be more effective than each single-modality attack of the same size. Moreover, **we also report results about additional multi-modal attacks**.  First, we apply the Multi-modal Embedding Attack (MEA) method [1], where the co-attack is designed based on the alteration of the joint representation rather than the prediction. Second, we also apply two multi-modal attack methods, introducing Multi-modal Gaussian Noise (MGN) and randomly Multi-modal Pixel Missing (MPM).  These experiments indicate that except for existing multi-modal attacks (FGM, PGD-$\ell_2$), our method can also resist various types of multi-modal attacks.
>
> |                           |   JT   |    GB   |   OGM   |   PMR   |    AT   |   Mixup   |  MSEFM  | CRMT-JT (ours) | CRMT-Mix (ours)| CRMT-AT (ours) |
> |:-------------------------:|:-------:|:-------:|:-------:|:-------:|:-------:|:-------:|:-------:|:-------:|:--------:|:-------:|
> |     MGN     | 0.5254  | 0.4622  | 0.5269  | 0.4797  | 0.5240  | 0.5015  | 0.4775  | 0.5603  |  0.6337  | 0.6039  |
> | MPM | 0.3154  | 0.3445  | 0.4462  | 0.3278  | 0.3699  | 0.4673  | 0.2980  | 0.5073  |  0.5698  | 0.5480  |
> |            MEA            | 0.3401  | 0.4121  | 0.3626  | 0.4549  | 0.5879  | 0.4782  | 0.5000  | 0.5560  |  0.5654  | 0.6890  |
>
> [1] J. Zhang, Q. Yi, and J. Sang, “Towards adversarial attack on vision- language pre-training models,” in *Proceedings of the 30th ACM International Conference on Multimedia*, 2022, pp. 5005–5013.

---

### Official Review · Reviewer_zQSE · 2023-11-05

**Soundness:** 3 good
**Presentation:** 3 good
**Contribution:** 4 excellent
**Rating:** 6
**Confidence:** 3

**Summary:**

The paper studies adversarial robustness for multi-modal learning by building a new lower bound for the perturbation radius through uni-modal margins and the Lipschitz constraint. Based on the proposed lower bound, a two-step adversarial training framework has been provided to improve the robustness of multi-modal learning. Experimental results on three benchmark datasets were provided regarding multiple attack methods, compared with several strong baselines.

**Strengths:**

- **New findings**: While discussing a new lower bound with the Lipschitz constraint is nothing new for adversarial robustness, the proposed method provides theoretical and insightful analyses of how the attack on a preferred modality would impact the overall robustness. This is a practical and common problem in multi-modal integration, as one modality often dominates the others.
- **Good presentation**: The reviewer enjoyed reading the presentation of the proposed method, where each step was well demonstrated with theoretical supports and clearly developed through proper treatments. One minor suggestion is to provide a pseudo `Algorithm` to outline the method, as well as add a `Remark` to better summarize and explain the training steps.
- **Decent experiment design**: Despite some minor issues, the experiment is overall well-designed and sufficient by 1) comparing with two groups of strong baselines, 2) adopting multiple attack methods (e.g., FGM, PGD, and missing modality), and 3) providing detailed ablation study and model discussions.

**Weaknesses:**

- **Missing implementation details**: I may have missed something; however, I did not find any implementation details about the multi-modal encoders. What are the backbones used in the experiment? Can the proposed method apply to different backbones?
- **Unclear model-specific weights/classifiers**: The exact role of introducing model-specific weights $a^{(m)}$ is somewhat unclear to me. How will it be used to guide the orthogonal classifier of each modality? Also, it remains unclear to me how the proposed eventually gets the prediction result upon different modalities's classifiers.
- **Lacking empirical evidence**: One main motivation of the proposed approach is one modality may be more vulnerable than the others.  While the adversarial accuracy (between uni-modal and multi-modal attacks) could support this observation empirically, it would be more convincing to provide more evidence that can be used to back-up the theoretical results, such as plotting the vulnerability indicator ($\eta$) values, visualizing the perturbation radius over modalities, etc.

**Questions:**

Please refer to the questions raised in the *Weakness* section. Plus, the reviewer is interested in the following questions:
- Can the proposed method apply to multiple modalities larger than 2?
- What's the selection criterion in choosing datasets for the experiment?
- Are the provided theoretical results applicable to vision-text data? Any empirical evidence?
- Could the proposed method be incorporated into the pre-trained multi-modal model (e.g., CLIP or BLIP)?

---

> ### Author Response · Authors · 2023-11-19
> **Response by authors**
>
> Thank you for your valuable feedback and insightful comments! We respond to some concerns below:
>
> 1. Missing implementation details: What are the **backbones** used in the experiment? **Apply to different backbones**?
>
> Thank you for your comments.
>
> a. As shown in Section 4.1 in the manuscript, we present that we use the widely used backbone ResNet18 as the uni-modal encoder, for Audio, Vision, and Optical Flow modalities. Then, the uni-modal representation is concatenated to form the multi-modal joint representation. We have updated the experiment section to clarify this setting.
>
> b. We conduct experiments on different backbones, including ResNet34 (Vision, V+Audio, A), and ResNet18 (Vision, V) + Transformer (Audio, A) on the Kinetic-Sounds dataset. These experiments show the improvement and the flexibility of our method across different backbones.
>
> |  ResNet34 (V)+ResNet34(A) |  Clean  | Missing #v | Missing #a |   FGM   |  PGD-$\ell_2$ |
> |:--------:|:-------:|:---------:|:---------:|:-------:|:-------:|
> | JT  | 0.6424  |  0.4528   |  0.2471   | 0.3132  | 0.2863  |
> |  CRMT-JT (ours) | 0.7435  |  0.5269   |  0.5705   | 0.4978  | 0.4746  |
>
> |  ResNet18(V)+Transformer(A)  |  Clean  | Missing #v | Missing #a |   FGM   |  PGD-$\ell_2$ |
> |:--------:|:-------:|:---------:|:---------:|:-------:|:-------:|
> | JT | 0.5538  |  0.3387   |  0.2703   | 0.2456  | 0.2100  |
> |  CRMT-JT(ours) | 0.5807  |  0.3721   |  0.3539   | 0.3067  | 0.2711  |
>
>
> 2. The exact **role of introducing model-specific weights $a$** is somewhat unclear to me. How will it be used to guide the orthogonal classifier of each modality? Also, it remains unclear to me how the proposed eventually **gets the prediction** result upon different modalities' classifiers.
>
> Thank you for your comments. The modality-specific weights can help the model identify the importance of each modality. As shown in Equation 8:
>
> \begin{equation}
> \begin{aligned}
>          \tilde{h}_k(\boldsymbol{x}) =  a_k^{(1)} \tilde{W} _{k\cdot}^{(1)} \phi^{(1)} (\boldsymbol{x}^{(1)}) +  a_k^{(2)}\tilde{W} _{k\cdot}^{(2)} \phi^{(2)} (\boldsymbol{x}^{(2)}) + \tilde{b}_k,
> \end{aligned}
> \end{equation}
> The $\tilde{h} _k(\boldsymbol{x})$ denote the logits score of $k$-th class. Since the valuable information between modalities is different, the weight $\boldsymbol{a}^{(m)}$ can lead the model to focus on the more reliable modalities. Integrating these weights can enhance the model's ability to effectively utilize the most valuable information from each modality. And the orthogonal $\tilde{W} _{k\cdot}^{(m)}$ for each modality, is approximated through the weight normalization method [1]. After we obtain the classifier, we can further get the logits score using the equation above and further obtain the prediction result. We have improved our presentation in Section 3.4 to explain the design more precisely.
>
> [1] L. Huang, X. Liu, B. Lang, A. Yu, Y. Wang, and B. Li, “Orthogonal weight normalization: Solution to optimization over multiple dependent stiefel manifolds in deep neural networks,” in *Proceedings of the AAAI Conference on Artificial Intelligence*, vol. 32, no. 1, 2018.
>
> 3. Evidences that **back-up the theoretical results**, such as plotting the vulnerability indicator (eta) values, and visualizing the perturbation radius over modalities.
>
> Thank you for your suggestions. As shown in Figure 3 in the manuscript, we demonstrate how the ratio of the vulnerability indicator ($\eta$) varies in our method. As shown in Figure 3(a), in AT method, there is also a large imbalance on vulnerable indicators, which is alleviated by our proposed CRMT method. Furthermore, to clearly explain this phenomenon, in revised Section B.4.1, we provide the heat map of the indicator $\eta$ to represent the robustness of each uni-modality, where the smaller indicator means the modality is more robust. Meanwhile, we also provide the uni-modal perturbation radius to further verify this modality preference, where we list the percentage of safe samples that can always resist this size of perturbation. It can be seen that in the Kinetic-Sounds dataset, the audio modality is definitely more vulnerable than the vision modality, thus explaining the phenomenon in Figure 1 in the manuscript.
>
> | Perturbation radius | 0.25    | 0.50     | 0.75    | 1.00       | 1.25    | 1.50     |
> |---------------------|---------|---------|---------|---------|---------|---------|
> |  Percentage of safe samples #v                 | 0.6366  | 0.6279  | 0.6214  | 0.6163  | 0.6068  | 0.6010  |
> | Percentage of safe samples #a                  | 0.4172  | 0.2573  | 0.1584  | 0.0996  | 0.0596  | 0.0422  |

---

> ### Author Response · Authors · 2023-11-19
> **Response by authors**
>
> 4. Apply method to **multiple modalities larger than 2**.
>
> Thank you for your suggestion. We apply the experiments on UCF101 with three modalities (RGB, Optical Flow (OF), and RGB Frame Difference (FD)), with two training strategies, training from scratch and with ImageNet-pretrained ResNet18. The following results show that our method can be well extended to multiple modalities.
>
> |   UCF101 (RGB+OF+FD)    | JT (Scratch) | CRMT-JT (Scratch, ours) | JT (Pretrained) | CRMT-JT (Pretrained, ours) |
> |:----------:|:--------------------------:|:-------------------------------:|:---------------:|:--------------------:|
> |    Clean   |           0.4490           |             0.5640              |      0.8312     |        0.8506        |
> |     FGM    |           0.4005           |             0.4567              |      0.2471     |        0.4138        |
> |   PGD_$\ell_2$   |           0.3963           |             0.4312              |      0.0783     |        0.2623        |
>
>
> 5. The **selection criterion** in **choosing datasets** for the experiment.
>
> In this study, we analyze the robustness through the perturbation radius, where the perturbation inside this radius can be always defended. Hence, our methods have no special requirements for certain modalities. Following previous studies in multi-modal learning [1,2], we apply the universally used Kinetic-Sounds, UCF101, and VGG-Sounds datasets. Our method can also extend to more modalities like the text to enhance the robustness.
>
> [1] J. Chen and C. M. Ho, “Mm-vit: Multi-modal video transformer for compressed video action recognition,” in *Proceedings of the IEEE/CVF winter conference on applications of computer vision*, 2022, pp. 1910–1921.
>
> [2] X. Peng, Y. Wei, A. Deng, D. Wang, and D. Hu, “Balanced multimodal learning via on-the-fly gradient modulation,” in *Proceedings of the IEEE/CVF Conference on Computer Vision and Pattern Recognition*, 2022, pp. 8238–8247.
>
>
> 6. Are the provided theoretical results applicable to **vision-text data**? Any empirical evidence?
>
> Thank you for your suggestion. We apply experiments on a vision-text dataset Food101, which inputs an image-text pair for classification. We employ a Vision Transformer (ViT) as our image encoder and BERT as our text encoder, subsequently concatenating their outputs to achieve a unified joint representation. To evaluate robustness, we apply multiple attacks including modality missing and descent-based attacks (FGM and PGD-$\ell_2$). It is important to note that attacks like the Fast Gradient Method (FGM) and Projected Gradient Descent with $\ell_2$ norm (PGD-$\ell_2$) are typically applied to continuous data. Given that text represents discontinuous data, we focus on implementing these attack methods on the image modality. Our results reveal that the text modality is more critical than the image modality, as its absence significantly impacts model performance. Concentrating on the $\ell_2$-norm, we achieve enhanced robustness under both FGM and PGD-$\ell_2$ attacks. Our method demonstrates a notable performance increase in scenarios where text is absent, though there is a slight decline in performance when the image modality is missing. This could be attributed to the huge performance difference among text and image. This is a very interesting and challenging issue in building a robust model for both modalities, which we will focus on in the upcoming work. Thanks again for such a valuable suggestion!
>
>
> | Food101 |  Clean  | FGM on Image | PGD-$\ell_2$ on Image | Missing Text | Missing Image |
> |:-------:|:-------:|:------------:|:---------------------:|:------------:|:-------------:|
> |    JT   | 0.8218  |    0.7603    |        0.7281         |    0.0497    |    0.7831     |
> | CRMT-JT | 0.8257  |    0.7656    |        0.7313         |    0.0759    |    0.7779     |
>
> 7. Could the proposed method be **incorporated into the pretrained multi-modal model** (e.g., CLIP or BLIP)?
>
> CLIP and BLIP stand out as highly effective multi-modal models, with their robustness gaining considerable attention in recent studies. Notably, the CLIP models also prefer certain modalities, such as text [1]. Hence, it is expected to first analyze the potential influence of such preference within CLIP with our proposed approach. However, there is a slight difference in that CLIP (also BLIP) employs contrastive loss across modalities instead of discriminative learning over joint representation. In future work, we will attempt to theoretically analyze the potential influence of this, thanks again for the precious suggestion, which undoubtedly helps to improve the potential value of this work.
>
> [1] Z. Liu, C. Xiong, Y. Lv, Z. Liu, and G. Yu, “Universal vision-language dense retrieval: Learning a unified representation space for multi-modal retrieval,” in *The Eleventh International Conference on Learning Representations*, 2022.

---

### Official Review · Reviewer_yeWY · 2023-11-05

**Soundness:** 3 good
**Presentation:** 3 good
**Contribution:** 2 fair
**Rating:** 6
**Confidence:** 4

**Summary:**

In this paper, the authors tackle the challenge of improving the robustness of multi-modal models against perturbations, such as uni-modal attacks and missing modalities. They provide valuable theoretical insight, emphasizing the importance of larger uni-modal representation margins and reliable integration for modalities in achieving higher robustness. They introduce a training procedure, Certifiable Robust Multi-modal Training (CRMT), which effectively addresses modality preference imbalances and enhances multi-modal model robustness. Experimental results validate the superiority of CRMT in comparison to existing methods, demonstrating its versatility and effectiveness. Overall, this paper contributes to the field by providing a theoretical foundation and a practical method for enhancing the robustness of multi-modal models.

**Strengths:**

Overall, the paper advances multi-modal robustness understanding and presents a practical solution in CRMT with strong empirical results, and the potential for broader applications in the ML/Multimodal community.
- The paper offers a fresh perspective on multi-modal robustness, emphasizing the importance of larger uni-modal representation margins and reliable integration within a joint multi-modal framework.
- The research is methodologically sound, with well-designed experiments and clear presentation.
- The authors effectively communicate complex concepts, enhancing accessibility.

**Weaknesses:**

The paper included some results on transformer as fusion models, particularlly the Multi-Modal Transformer-based framework with hierarchical attention on the VGGS dataset. However, all experriments, especailly the one with transformerr adopt training from scratch and did not consider any pre-training strategies, such as uni-modal pretraining, or multi-modal pretraining. It will be interesting to consider such methods as baselines and also to see how much CRMT can help to improve.

Also, except for experimenting, it will be good if authors can discuss how does their method generalize to other fusion mechanisms, besides late fusion.

**Questions:**

N/A

---

> ### Author Response · Authors · 2023-11-19
> **Response by authors**
>
> Thanks for your valuable feedback and insightful comments! We respond to some concerns below:
>
> 1. Transformer adopted training from scratch and did not consider any **pretraining strategies**.
>
> Thank you for your advice on our work. To validate it, we conduct experiments using the ImageNet-pretrained Transformer on the Kinetic-Sounds dataset. Experimental results demonstrate that our method can also perform well with pretraining.
>
> | Transformer |  clean  |   FGM   |  PGD-$\ell_2$ |
> |:-----------:|:-------:|:-------:|:-------:|
> |   Baseline  | 0.6788  | 0.1366  | 0.0865  |
> |   CRMT (ours)   | 0.7406  | 0.3198  | 0.2078  |
>
> 2. How does the method **generalize to other fusion mechanisms**, besides late fusion.
>
> Thank you for your comments. In this manuscript, our analysis mainly focuses on the representative fusion strategy, late fusion, which is widely used in multi-modal research [1,2].
> Meanwhile, our method are adaptable to other fusion mechanisms, where the modality-specific representations could interact earlier in the process. Previously, we defined the margin as $\zeta^{(m)}_j(\boldsymbol{x}^{(m)})$, where $j$ is the nearest class to calculate the margin, and the margin is only determined by uni-modal input $\boldsymbol{x}^{(m)}$. To adapt our method for these scenarios including intermediate fusion, we can redefine the representation margin as $\zeta^{(m)}_j(\boldsymbol{x}^{(1)}, \boldsymbol{x}^{(2)})$, indicating that both modalities' input influence the margin. This modification allows us to extend the framework to measure multi-modal perturbations in a more integrated manner. Additionally, we can adapt the definition of the Lipschitz constant in our theoretical analysis here to measure how the multi-modal perturbation influences the margin:
>
> \begin{equation}
> \begin{aligned}
> |\zeta_j^{(m)}(\boldsymbol{x}^{(1)}, \boldsymbol{x}^{(2)}) - \zeta_j^{(m)}(\boldsymbol{x}'^{(1)}, \boldsymbol{x}'^{(2)})| \le \tau_j^{(m1)}\left\|\left\| \boldsymbol{x}^{(1)} - \boldsymbol{x}'^{(1)} \right\|\right\|_2 + \tau_j^{(m2)}\left\|\left\| \boldsymbol{x}^{(2)} - \boldsymbol{x}'^{(2)} \right\| \right\|_2
> \end{aligned}
> \end{equation}
> where $\tau_j^{(m1)}$ represents the Lipschitz constant of modality $m$ from modality $1$. This constant can reflect how the alteration in modality $1$ influences the margin in modality $m$. Then following the proof in the manuscript, we can observe that:
>
> \begin{equation}
> \begin{aligned}
>       c_j^{(1)} |\zeta_{j}^{(1)}(\boldsymbol{x}^{(1)}, \boldsymbol{x}^{(2)}) - \zeta_{j}^{(1)}(\boldsymbol{x}'^{(1)}, \boldsymbol{x}'^{(2)})|+ c_j^{(2)} |\zeta_{j}^{(2)}(\boldsymbol{x}^{(1)}, \boldsymbol{x}^{(2)}) - \zeta_{j}^{(2)}(\boldsymbol{x}'^{(1)}, \boldsymbol{x}'^{(2)})| \\\\         \leq  (c_j^{(1)} \tau_j^{(11)} + c_j^{(2)} \tau_j^{(21)})\left\|\left\|\boldsymbol{x}^{(1)} - \boldsymbol{x}'^{(1)}\right\|\right\|_2 + (c_j^{(1)} \tau_j^{(12)} + c_j^{(2)} \tau_j^{(22)})\left\|\left\|\boldsymbol{x}^{(2)} - \boldsymbol{x}'^{(2)}\right\|\right\|_2 .
> \end{aligned}
> \end{equation}
> Thus, we can obtain the perturbation bound in this setting:
>
> \begin{equation}
> \begin{aligned}
>  & \min_{\boldsymbol{x}'} \left\|\left\|\boldsymbol{x} - \boldsymbol{x}'\right\|\right\|_2
>  \geq \frac{c_j^{(1)} \zeta_j^{(1)}(\boldsymbol{x}^{(1)}, \boldsymbol{x}^{(2)}) + c_j^{(2)} \zeta_j^{(2)}(\boldsymbol{x}^{(1)}, \boldsymbol{x}^{(2)})+ \beta_j}{\sqrt{(c_j^{(1)} \tau_j^{(11)} + c_j^{(2)} \tau_j^{(21)})^2 +(c_j^{(1)} \tau_j^{(12)} + c_j^{(2)} \tau_j^{(22)})^2 }} \\\\
>   whe&re   \quad j \neq y \quad s.t. \quad  c_j^{(1)} \zeta_j^{(1)}(\boldsymbol{x}'^{(1)}, \boldsymbol{x}'^{(2)}) + c_j^{(2)} \zeta_j^{(2)}(\boldsymbol{x}'^{(1)}, \boldsymbol{x}'^{(2)}) + \beta_j = 0.
> \end{aligned}
> \end{equation}
>
> The idea of the proof is similar to the one in Section A.1 in the Appendix.
>
> Additionally, we present an experiment where our method is applied to intermediate fusion using the widely adopted MultiModal Transfer Module (MMTM) [3] on the Kinetic-Sounds dataset. The results demonstrate that our approach can enhance the robustness of intermediate fusion mechanisms.
>
> |   Kinetic-Sounds   |  Clean  | Missing #v | Missing #a |   FGM   |  PGD-$\ell_2$ |
> |:--------:|:-------:|:---------:|:---------:|:-------:|:-------:|
> | MMTM | 0.6693  |  0.4542   |  0.2028   | 0.3438  | 0.3205  |
> |   CRMT-MMTM (ours)   | 0.6737  |  0.5211   |  0.3169   | 0.3445  | 0.3358  |
>
>
> [1] Y. Huang, J. Lin, C. Zhou, H. Yang, and L. Huang, “Modality competition: What makes joint training of multi-modal network fail in deep learning?(provably),” *arXiv preprint arXiv:2203.12221*, 2022.
>
> [2] Y. Huang, C. Du, Z. Xue, X. Chen, H. Zhao, and L. Huang, “What makes multi-modal learning better than single (provably),” *Advances in Neural Information Processing Systems*, vol. 34, 2021.
>
> [3] H. R. Vaezi Joze, A. Shaban, M. L. Iuzzolino, and K. Koishida, “Mmtm: Multimodal transfer module for cnn fusion,” in *Conference on Computer Vision and Pattern Recognition* (CVPR), 2020.

---

### Author Response · Authors · 2023-11-19
**Thanks for reviewing our work**

Dear all reviewers, we sincerely appreciate your encouraging and positive feedback on our work. We are glad that you recognize our study as offering a new perspective and value our theoretical and empirical contributions. In response to your insightful comments and valuable suggestions, we have dedicated considerable effort over the past few days to integrate your advice into our work. We have accordingly revised our submission to reflect these changes.

In short, we aim to summarize the key modifications we have made to our manuscript:

**Experiment**:
we have conducted extensive experiments detailed in the Appendix. Specifically, in Section B.2 of the revised Appendix, we explore variations involving different backbone architectures, fusion techniques, and comparative analyses with recent robust methods. This section also includes experiments on additional datasets comprising three modalities and image-text pairings. Furthermore, the revised Section B.3.1 provides a detailed examination of the vulnerability of certain modalities.

**Extensive Analysis**:
We have demonstrated the scalability of our method to scenarios involving more than two modalities, and with different fusion strategies, which are detailed in the revised Section A.4.

**Writing**:  We have made specific changes to unclear elaboration and update up-to-date works related to multi-modal robustness.

Thanks for your precious reviews.

---

### Meta-Review · Area_Chair_runM · 2023-12-07

**Metareview:**

This study delves into the robustness of multi-modal models when faced with perturbations, including uni-modal attacks and missing modalities. It offers valuable theoretical insights, underscoring the significance of broader individual modality representation margins and dependable integration for achieving heightened robustness. Introducing the Certifiable Robust Multi-modal Training (CRMT) method, the researchers effectively address imbalances in modality preferences, bolstering the robustness of multi-modal models.

Strengths:
As highlighted by the majority of reviewers, this work introduces a fresh perspective on multi-modal robustness, emphasizing the importance of wider uni-modal representation margins and dependable integration within a unified multi-modal framework. Moreover, the study demonstrates methodological rigor with a clear presentation.

Weakness:
Initial reviews raised several concerns regarding the experimental outcomes. While the authors successfully addressed most of these concerns during the rebuttal, certain issues remain. For instance, the absence of robustness testing against the strongest adversarial attacks, such as Auto Attacks, raises the possibility of gradient masking in the reported attacks. Additionally, the study's generalizability to CLIP models trained with contrastive loss remains unexplored.

Based on unanimous agreement among reviewers, we recommend accepting this work, acknowledging its valuable contributions despite the outlined concerns.

**Justification For Why Not Higher Score:**

Although the authors have addressed most of the concerns raised by reviewers, there are remaining concerns, e.g., the evaluated attacks are not the strongest Auto Attacks, and the methods have not been generalized to state-of-the-art CLIP models.

**Justification For Why Not Lower Score:**

Based on the scientific merits of this work and the strengths outlined above, all reviewers agree to accept this work.

---

### Decision · Program_Chairs · 2024-01-16

Accept (poster)